# The Efficacy of Early Interventions for Children with Autism Spectrum Disorders: A Systematic Review and Meta-Analysis

**DOI:** 10.3390/jcm11175100

**Published:** 2022-08-30

**Authors:** Sofia Daniolou, Nikolaos Pandis, Hansjörg Znoj

**Affiliations:** 1Department of Psychology, University of Bern, 3012 Bern, Switzerland; 2Department of Orthodontics and Dentofacial Orthopedics, University of Bern, 3010 Bern, Switzerland

**Keywords:** autism spectrum disorders, early interventions, cognitive ability, language, meta-analysis

## Abstract

The superiority of early interventions for children with autism spectrum disorders (ASDs) compared to treatment as usual (TAU) has recently been questioned. This study was aimed to investigate the efficacy of early interventions in improving the cognitive ability, language, and adaptive behavior of pre-school children with ASDs through a systematic review of randomized controlled trials (RCTs). In total, 33 RCTs were included in the meta-analysis using the random effects model. The total sample consisted of 2581 children (age range: 12–132 months). Early interventions led to positive outcomes for cognitive ability (*g* = 0.32; 95% CI: 0.05, 0.58; *p* = 0.02), daily living skills (*g* = 0.35; 95% CI: 0.08, 0.63; *p* = 0.01), and motor skills (*g* = 0.39; 95% CI: 0.16, 0.62; *p* = 0.001), while no positive outcomes were found for the remaining variables. However, when studies without the blinding of outcome assessment were excluded, positive outcomes of early interventions only remained for daily living skills (*g* = 0.28; 95% CI: 0.04, 0.52; *p* = 0.02) and motor skills (*g* = 0.40; 95% CI: 0.11, 0.69; *p* = 0.007). Although early intervention might not have positive impacts on children with ASDs for several outcomes compared to controls, these results should be interpreted with caution considering the great variability in participant and intervention characteristics.

## 1. Introduction

The increasing prevalence rates in autism diagnoses during recent years (1 in every 150 children in 2000, 1 in every 68 children in 2012, and now 1 in every 44 children) [1] has enhanced the establishment of a variety of interventions for young children with ASDs [2,3,4,5,6]. Such approaches are classified according to their manual and targeted outcomes as behavioral interventions, developmental interventions, naturalistic developmental behavioral interventions (NCBI), TEACCH, sensory-based interventions, animal-assisted interventions and technology-based interventions [2]. Sensory-based interventions are motivated by the theory that children with ASDs may fail to respond to sensory inputs such as sound, touch, body movement, sight, taste, and smell. Within this concept, sensory integration therapy aims to help children with ASDs use their senses together to enhance their engagement and participation in a range of daily living activities. For example, sensory stimuli, such as a hug machine with deep pressure, can provide a calming effect and reduce unwanted movements in children with ASDs during travelling [7].

Despite the existence of various treatment programs, there is insufficient evidence for the superiority of a treatment model in improving core areas of deficits of children with ASDs, such as cognitive ability, language, communication, socialization and adaptive behavior. Recently, the American Psychological Association (APA) published a meta-analysis about the efficacy of early interventions. The authors concluded that, when no study quality criteria were considered, positive outcomes were found for behavioral, developmental and NCB interventions. However, when the analysis was limited to RCTs at a low risk of detection bias, there was no evidence of positive outcomes for young children with ASDs [2]. Partially consistent with the results published by APA is the meta-analysis conducted by Yi et al. (2019) [8]. The authors compared the effectiveness of applied behavior analysis (ABA), early start Denver model (ESDM), Picture Exchange Communication Systems (PECS), and discrete trial training (DTT) investigated via randomized controlled trials (RCTs). They found positive outcomes of ABA-based interventions in socialization, communication, and expressive language, but not in receptive language, adaptive behavior and cognitive ability. Rogers et al. (2021) [9], conducted a meta-analysis of non-randomized studies about the efficacy of ABA. They did not find any significant outcomes for cognitive ability or adaptive behavior, but children in the experimental group outperformed children in the control group in the adaptive behavior scale over a 2-year follow-up. Positive outcomes for children in cognitive competence, language-communication, social competence, and adaptive behavior were also reported in a substantial number of meta-analyses [10,11,12,13,14,15,16,17,18]. However, the aforementioned findings were not ubiquitous [19,20,21].

The majority of previous meta-analyses pointed out that the considered studies had considerable methodological limitations [1,18,22,23]. Moreover, if we carefully examine the studies testing the effectiveness of early interventions, we notice that regardless of their theoretical frameworks or “brand-name”, they presented extensive differences in terms of treatment intensity and duration. Some studies included more comprehensive interventions, focusing on core functional areas such as cognitive ability, language, or adaptive behavior, while others included targeted interventions addressing more restricted areas, such as joint attention and imitation. However, children with autism who present higher scores of joint attention, imitation, and object play in infancy are more likely to have stronger communication and intellectual skills in the subsequent years [24].

The aim of the current meta-analysis was to draw a valid conclusion about the efficacy of early intervention programs for pre-school children with ASDs compared to children that did not receive any of the abovementioned early intervention treatments in improving their cognitive ability, language skills, communication, socialization and adaptive behavior. Furthermore, this is the first meta-analysis to synthesize all available information from the included studies through mathematical formulas that enabled the combined testing of multiple measurements of the dependent variables across studies (e.g., single variables measured by different scales). Considering the impact that the improvement in proximal variables can have on distal variables [24], we included studies focusing on both comprehensive and targeted areas of functionality. We also aimed to extend the findings of Sandback et al., 2020 [2] and examine whether intervention duration and intensity can predict the performance of participants post treatment. To fulfil this purpose, we decided to only include RCTs, because this design can provide more reliable results regarding the causal effect of an intervention [25].

## 2. Materials and Methods

### 2.1. Search Strategy

We searched PsycInfo, ERIC and MEDLINE PubMed, and Google Scholar on 11 May 2022 without applying any time limit for peer-reviewed studies published in English language by entering the following keywords: (autism) OR (autistic) OR (developmental disorder) OR (autism spectrum disorder) OR (Asperger)) AND ((preschool age children) OR (young children) OR (toddlers) OR (pupils)) AND ((intervention) OR (comprehensive intervention) OR (parent training) OR (parent implemented) OR (comprehensive approach) OR (developmental approach) OR (behavioral approach) OR (therapy) OR (EIBI) OR (ABA)) AND ((cognition) OR (cognitive ability) OR (language) OR (adaptive behavio*)). A pilot search was first conducted in October 2021. The literature search was performed by two independent authors. Disagreements were solved after discussion and the re-evaluation of the relevance of each study until a consensus was reached. 

Query logic was adapted to each search database to optimize retrieval. Following the recommendations by [26], the study selection process was conducted and presented using the Preferred Reporting Items for Systematic Reviews and Meta-Analyses (http://prisma-statement.org/, accessed on 26 May 2022) as a guide (see Figure 1). The PRISMA study selection process entails four phases: identification, screening, eligibility and final synthesis.

### 2.2. Study Selection

We included studies that: (i) were randomized controlled trials (RCTs); (ii) focused on participants who were infants at risk of autism and preschool-aged children with a diagnosis of autistic spectrum disorders (ASDs), autistic disorder (AD), pervasive developmental disorders-not-otherwise-specified (PDD-NOS), and/or pervasive developmental disorders (PDDs) (studies including exclusively infants who were all under 18 months of age were not included, since we aimed at measuring a clear manifestation of the autistic or developmental symptoms); (iii) considered interventions that included psychosocial parent- or/and professional-implemented specialized interventions aiming at reducing ASD-related impairments (studies examining pharmacological treatments and alternative interventions such as music therapy or equine therapy were not included, and since we tested the overall effectiveness of early interventions, we did not include studies examining the effectiveness of different versions of the same approach, varying in intervention providers, setting, or dosage, for example, or studies comparing different kinds of early interventions); (iv) measured outcomes that included at least one of the following domains—cognitive ability, expressive language, receptive language, communication, socialization, adaptive behavior composite, daily living skills and motor skills—measured by standardized scales (outcomes had to be presented by means and standard deviations for both groups); and (v) were published in the English language and were peer-reviewed articles.

### 2.3. Data Extraction and Coding

We created three different spreadsheets in Excel (version 2206, Microsoft Excel, Microsoft Corporation, Redmont, WA, USA)**.** The first spreadsheet contained descriptive information of each study including study name, date, study sample size, participant’s age, gender, and intervention-related information such as intervention type, duration, intensity, intervention providers, and intervention setting (Table 1). Experimental group interventions included any intervention. The intervention duration and intensity were coded as moderators of the result and were thus analyzed as independent variables. The second spreadsheet contained the means and standard deviations, as well as the scales of measurement of the examined variables for each outcome, and the third spreadsheet was identical to second for the combined outcomes (Table A1 and Table A2).

### 2.4. Risk of Bias

The quality of each study was assessed with the Cochrane risk of bias tool RoB 2 [60] by two independent examiners. This tool includes six items that cover the following bias domains: (i) bias arising from the randomization process, (ii) bias due to deviation from intended interventions, (iii) bias due to missing outcome data, (iv) bias in the measurement of the outcome, (v) bias in the selection of the reported results, and (vi) overall bias. This tool has three grading levels: (i) low, (ii) unclear, and (iii) high risk of bias. The worst grading in individual items define the overall risk of bias for each single study.

### 2.5. Data Analysis

Meta-analysis was performed using the Review Manager (RevMan, version 5.4.1, The Nordic Cochrane Centre, Copenhagen, Denmark). RevMan is the Cochrane Collaboration’s software for preparing and maintaining Cochrane reviews. Because there was meaningful variability across studies regarding participant and intervention characteristics, we analyzed the results using the random effects model of meta-analysis. In our study, we used variable instruments for assessing the same variable (e.g., expressive language was measured with Mullen Scales of Early Learning (MSEL), MacArthur Communicative Development Inventories (MCDI), or other standardized scales). We converted all the measurements to standardized mean differences and variances so that they could be comparable to each other. The standardized mean difference is the difference in mean outcome between groups divided by the standard deviation of outcome across participants [25]. Subsequently, based on the standardized mean differences and variances, we calculated Hedges’ g with RevMan software. Hedges’ g and Cohen’s d were interpreted in the same way according to the rule of thumb that Cohen suggested, where an effect size of 0.20 is small, an effect size of 0.50 is moderate, and an effect size of 0.80 is large [61].

After computing the effect sizes and their statistical significance, we conducted a heterogeneity test in order to establish whether our data were consistent. The heterogeneity was assessed using tau^2^, a metric that we used to define the variance of the true effects sizes and to determine the weight assigned to each included study analyzed with the random effects model [62]. Additionally, we calculated the *I*^2^ statistic, which describes the magnitude of heterogeneity across studies that is attributable to the true differences of the results rather than chance or sampling error [63]. Heterogeneity can be interpreted as low when *I*^2^ =0–40%, as moderate when *I*^2^ = 30–60%, as substantial when *I*^2^ = 50–90% and as considerable when *I*^2^ = 75–100% [62].

In the current review, many of the included studies contained outcomes that were measured by more than one scale (MSEL and MCDI for expressive language). We could not analyze the different outcomes as they were independent because this could lead to incorrect estimates of the variance for the summary effect [62]. Since we analyzed the standardized mean differences for each outcome, we calculated an effect size for all the multiple outcomes per variable for each study. In this case, we calculated the mean effect sizes and the variances for all the multiple outcomes and then the corresponding standard errors (SEs). To compute the combined variance, we applied the formulas suggested by Borenstein et al., (2021) [62]. In this way, we could include all the relevant and available information across studies and at the same time address the problem of non-independence, since all the measurements per study came from the same sample. The formulas were as follows.

(1)Computed variance in case we had two outcomes per study.


$$V_{\bar{Y}}=1/4\left(V_{Y1}+V_{Y2}+2r\sqrt{V_{Y1}}\;\sqrt{V_{Y2}}\right)$$


(2)Computed variance in case we had more than two outcomes per study.


$$V_{\bar{Y}}={\left(\frac{1}{m}\right)}^{2}var\left(\overset{m}{\underset{j=1}{\sum }}Yi\right)={\left(\frac{1}{m}\right)}^{2}\left(\overset{m}{\underset{j=1}{\sum }}V_{i}+\underset{j\neq k}{\sum }\left(r_{jk}\sqrt{V_{j}}\sqrt{V_{k}}\right)\right)$$


## 3. Results

### 3.1. Search Results

A PRISMA flowchart summarizing the article selection process is presented in Figure 1. After the initial database search, 5057 studies, plus two studies identified in Google Scholar, were retrieved. After excluding duplicates and studies that did not meet our inclusion criteria, 33 studies were included in the analysis.

### 3.2. Study Characteristics

A full description of the included studies [27,28,29,30,31,32,33,34,35,36,37,38,39,40,41,42,43,44,45,46,47,48,49,50,51,52,53,54,55,56,57,58,59] is depicted in Table 1. The total number of children was 2581. All children had a diagnosis of either ASDs or PDD. The participant age at the beginning of the study ranged from 12 to 132 months.

Out of the 33 studies included, 12 studies were categorized as long-term interventions, 9 were categorized as medium-term interventions, and 12 were categorized as short-term interventions. Additionally, 10 studies implemented high-intensity interventions and 23 studies implemented low-intensity interventions. Twenty-two studies compared the interventions of our interest with TAU, three studies compared early interventions with no treatment or a WL, and eight studies included an altered or low intensity intervention compared to the intervention of the experimental group. The duration of the provided interventions ranged from 12 weeks to 2 years, and their intensity ranged from 3 h of parent sessions every 6 weeks to 15.2 h of therapist-delivered and 16.3 h per week of parent-delivered therapy 5 days per week for 2 years. Intervention providers were both professionals and parents in 13 studies, parents only in 17 studies, and professionals only in three studies. In 27 studies, the intervention setting was individual therapy; in two studies, the setting was both group and individual therapy; and in four studies, the setting was group therapy. All the studies included in meta-analysis reported results obtained from standardized tests.

### 3.3. Risk of Bias Assessment

All studies were assessed for risk of bias by two independent authors (Figure 2). Disagreements were solved through the re-evaluation of the original papers and discussion until a consensus was reached. In general, 11 studies were assessed as having some concerns due to a lack of specific information and three studies were assessed as having a high risk of bias in randomization process criterium. All studies were assessed as having a high risk of bias in deviation from intended intervention criterium because the participants and personnel were not blinded to intervention status. Four studies were assessed as having some concerns, and seven studies were assessed as having a high risk of bias due to missing outcome data criterium. Two studies were assessed as having some concerns, and five studies were assessed as having a high risk of bias in the measurement of the outcome criterium. Finally, two studies were assessed as having some concerns regarding bias in the selection of the reported results.

After the completion of the risk of bias assessment, studies that were assessed as having a high risk of bias in the measurement of the outcome criterium were excluded from the analysis.

### 3.4. Meta-Analysis

Appendix A (Table A1) contains the pre- and post-measurements for every variable across studies, and (Table A2) contains tables with the effect sizes of the combined outcomes after the statistical formulas were applied.

#### 3.4.1. Sensitivity Analysis

In order to calculate the variance for the combined outcomes, we had to provide their correlation coefficients. Since we did not know the correlation between our combined outcomes, we assumed it was *r* = 0.5. Subsequently, we performed a sensitivity analysis for *r* = 0.25 and *r* = 0.75, and the results confirmed our assumption, since we did not observe any differences in the results.

#### 3.4.2. Cognitive Ability Results

The overall effect size of cognitive ability was based on data from 12 studies (Figure 3). The overall result of the meta-analysis indicated that early intervention programs are efficacious in improving the cognitive ability of pre-school children with ASDs (*g* = 0.32; 95% CI: 0.05, 0.58; *p* = 0.02) based on the pre-treatment and post-treatment assessments.

A subgroup analysis was performed to test whether intervention intensity and intervention duration modified the effect of early intervention in comparison to control conditions (analysis not presented). However, the number of trials and participants contributing data to the intervention duration subgroups (5 trials and 543 participants for long-term interventions, 3 trials and 120 participants for medium-term interventions, and 4 trials and 214 participants for short-term interventions) and the intervention intensity subgroups (7 trials and 614 participants for high-intensity interventions and 5 trials and 263 participants for low-intensity interventions) was unequal, meaning that the analysis was unlikely to produce useful findings [64].

After the exclusion of studies with bias in measuring of the outcomes, there were no positive outcomes of early intervention for cognitive ability (*g* = 0.25; 95% CI: −0.04, 0.54; *p* = 0.09).

#### 3.4.3. Language Results

The analysis was based on 26 studies for expressive language and on 23 studies for receptive language (Figure 4 and Figure 5). After combining the results of studies with multiple outcomes, analysis showed that early interventions were marginally insignificant for expressive language (*g* = 0.10; 95% CI: −0.00, 0.20; *p* = 0.06) and not efficacious in improving the receptive language skills of pre-school children with ASDs (*g* = 0.12; 95% CI: −0.06, 0.31; *p* = 0.19). The I^2^-statistic showed that the heterogeneity among studies was insignificant for expressive language (*Tau^2^* = 0.01; *I*^2^ =20%, *p* = 0.18) and substantial and significant for receptive language (*Tau^2^* = 0.14; *I*^2^ =74%, *p* ≤ 0.0001).

The test for subgroup differences indicated that there was no statistically significant subgroup effect (*p* = 0.16 for expressive language and *p* = 0.09 for receptive language; analysis not presented), suggesting that intervention duration does not modify the effect of early intervention in comparison to control conditions. However, the number of trials and participants that contributed data to the subgroups was unequal, meaning that the analysis may not have been able to detect subgroup differences. Additionally, although the subgroup analysis performed to test whether intervention intensity modifies the effect of early intervention in comparison to control conditions (analysis not presented) indicated that there was a statistically significant subgroup effect, the number of trials and participants that contributed data to the intervention intensity subgroups (8 trials and 613 participants for high-intensity interventions and 18 trials and 1278 participants for low-intensity interventions for expressive language; 6 trials and 522 participants for high-intensity interventions and 17 trials and 1208 participants for low-intensity interventions for receptive language) was unequal, meaning that the analysis was also unlikely to produce useful findings [64].

However, when studies with bias in the measurement of the outcome were excluded from the analysis, the results remained insignificant for expressive language (*g* = 0.07; 95% CI: −0.04, 0.18; *p* = 0.20).

#### 3.4.4. Adaptive Behavior Composite, Communication, Socialization, Daily Living Skills and Motor Skills Results

The final effect size for the adaptive behavior composite result was based on the results from seven studies (Figure 6) and showed that early intervention was not effective for the adaptive behavior composite (*g* = 0.20; 95% CI: −0.16, 0.55; *p* = 0.27). Analysis also indicated that early interventions were not statistically significant, either for improving communication (17 studies, *g* = 0.06; 95% CI: −0.07, 0.12; *p* = 0.36, Figure 7) and socialization (16 studies, *g* = 0.10; 95% CI: −0.06, 0.27; *p* = 0.21, Figure 8); on the other hand, early interventions were statistically significant for daily living (seven studies, *g* = 0.35; 95% CI: 0.08, 0.63; *p* = 0.01, Figure 9) and motor skills (sight studies, *g* = 0.39; 95% CI: 0.16, 0.62; *p* = 0.001, Figure 10).

The *I*^2^-statistics showed that the heterogeneity among studies was moderate for socialization (*Tau^2^* = 0.07; *I*^2^ = 55%, *p* = 0.004) and statistically insignificant for all the other variables. Non-significant heterogeneity tests for the remaining outcomes possibly occurred due to a low power, since the number of studies included in these analyses was small [25].

The test for subgroup differences indicated that there was no statistically significant subgroup effect for the adaptive behavior composite (*p* = 0.70), communication (*p* = 0.54), socialization (*p* = 0.51), daily living skills (*p* = 0.56), and motor skills (*p* = 0.27), suggesting that intervention duration does not modify the effect of early interventions in comparison to control conditions. However, the number of trials and participants that contributed data to the subgroups was unequal, meaning that the analysis may not have been able to detect subgroup differences. Similarly, although the subgroup analysis performed to test whether intervention intensity modified the effect of early intervention in comparison to control conditions (analysis not presented) indicated that there was a statistically significant subgroup effect for socialization, the number of trials and participants that contributed data to the intervention intensity subgroups (4 trials and 191 participants for high-intensity interventions and 12 trials and 754 participants for low-intensity interventions) was unequal, meaning that the analysis was also unlikely to produce useful findings [64].

After the exclusion of studies with a high risk of bias, results remained positive for daily living skills (*g* = 0.28; 95% CI: 0.04, 0.52; *p* = 0.02) and motor skills (*g* = 0.49; 95% CI: 0.28, 0.79; *p* ≤ 0.00001).

#### 3.4.5. Follow-Up Data

Since early intervention may initiate a cascade of developmental events that is not yet apparent at post-treatment [65], follow-up data recorded sometime after the intervention period is over could provide evidence of the efficacy of early interventions [2]. For this reason, we conducted a separate analysis of follow-up data despite the limited number of studies that reported follow-up data. These results must be interpreted with caution. A detailed depiction of follow-up data is shown in Figure 11. Overall, the analysis of follow-up data did not provide evidence of the sustainability of positive outcomes. A positive result was found for daily living skills (*g* = 0.46, *p* = 0.03), and a marginally insignificant result was found for the adaptive behavior composite (*g* = 0.34, *p* = −0.04). However, due to the instability of the estimated pooled effects, we recommend a descriptive use of Figure 11 regarding the results of the included studies.

### 3.5. Publication Bias

A possible way to assess publication bias is via a funnel plot [66]. Regarding publication bias, the results of smaller studies were spread widely, due to lower precision, and asymmetrically around the average estimate compared to the results of larger studies. This asymmetry is suggestive of missing studies. In the absence of publication bias, individual study results are more evenly distributed around a pooled estimate. However, caution should be exercised when interpreting funnel plots, especially when the number of included studies is smaller than 10 (ref. Cochrane handbook) [29,67]. In our case, the funnel plot (Figure 12) indicates no publication bias.

## 4. Discussion

The purpose of this meta-analysis was to assess the efficacy of early interventions in pre-school children with ASDs and investigate how the intervention intensity and duration could mediate the final outcome. Considering the overall effect, when all the included studies were analyzed, early interventions showed significant effects on the cognitive ability, daily living skills, and motor skills of children with autism, while there were no additional benefits for expressive language, receptive language, communication, socialization, and adaptive behavior compared to whatever other interventions were provided. It should be noted here that the results for expressive language were marginally insignificant and worth further investigation. Moreover, when studies with detection bias were excluded from the analyses, positive early intervention effects were detected solely for daily living skills and motor skills. The subgroup analyses on the intervention intensity and duration did not reveal any significant effects. This could be attributed to the fact that the number of studies and participants included in the subgroups were unequal, limiting the power to identify any actual effects of these moderators on the tested outcomes.

Apart from daily living skills and motor skills, our main results are consistent with the meta-analysis of Sandback et al., (2020) [2], where no positive effect of early intervention was detected when the analysis was based only on RCTs with no detection bias. Similarly, Rogers et al., (2021) [8], in their meta-analysis of non-randomized studies about the efficacy of ABA, did not find any significant effects on cognitive ability or adaptive behavior. Additionally, the results of our study are in partial agreement with those of Tachibana et al., (2017) [21], who also only included RCTs and excluded studies with a high risk of bias or studies that did not meet other decisive quality criteria from the final analysis. That study indicated that early intervention did not appear to be efficacious for expressive language, receptive language, and adaptive behavior. Howlin et al., (2009) [68] also failed to find a significant effect of EIBI for expressive language. Nevil et al., (2016) [20] also based their analysis on RCTs and reported that a parent-mediated intervention only resulted in minor improvements in socialization and cognition of children with ASDs. They also reported insignificant improvements in communication language, which incorporated the expressive and receptive language variables. In addition, a recent meta-analysis conducted by Reichow et al., (2018) [18] that examined the efficacy of EIBI concluded that there is low-quality evidence that EIBI can improve IQ, language and adaptive behavior. The authors also underlined the fact that most of the data were derived from studies with many methodological limitations that could have affected the outcomes. Additionally, Speckley et al., (2009) [19] examined the efficacy of applied behavioral intervention and concluded that it did not lead to significant improvements in cognitive ability, language and adaptive behavior compared to TAU. Furthermore, the results of our study are in partial agreement with those of the meta-analysis conducted by Peters-Scheffer et al., (2011) [69], who indicated that EIBI only showed statistically significant effects for IQ, not for expressive and receptive language or for communication, socialization and daily living skills. The described inconsistencies with previous studies could be attributed to the different inclusion criteria applied in each case, which resulted in considerably different study samples. They could also be partly or entirely attributed to biases within the included studies or to the quality of the collected data. In our study, only five of the included studies were assessed as having a low risk of bias in all criteria.

Various studies on the efficacy of early interventions have also demonstrated positive effects in specific areas. A series of meta-analyses that tested high-intensity interventions showed positive outcomes. Yu et al. (2020) [8] concluded that ABA-based interventions can have a positive effect on socialization, communication, and expressive language but not on receptive language, adaptive behavior, daily living skills, IQ, verbal IQ, nonverbal IQ, and cognition. Fuller et al., (2020) [70] conducted a meta-analysis regarding the efficacy of ESDM for children with ASDs and found positive outcomes for cognition and language but not for adaptive behavior. Finally, Warren et al., (2011) [71], Makrygianni et al., (2018) [12], and Reichow et al., (2011) [13] concluded that early intervention is efficient in improving the IQ, language, and adaptive behavior of children with ASDs.

Based on the current evidence, it is not yet clear whether early intervention is effective. There seems to be a trend where older meta-analyses more often reported positive outcomes while more recent ones that included studies of higher quality did not show such strong overall effects, only improvements on isolated variables if on any variable at all. Especially when only RCTs were included, the claimed positive effects could not be established. Considering the extended heterogeneity presented by children with ASDs and the different types of interventions tested, we cannot conclude at present that early intervention is not an effective intervention. There could be many explanations for the absence of effects. We suggest that the main priority in relevant research should be to identify which treatment works best and for whom [72,73]. It is not yet clear which characteristics of a child and family may affect the success of an intervention. Early interventions target children even at infancy and until more than ten years of age. In our review, the age range of the participants was found to be 24–132 months. Although there have been a limited number studies examining the influence of age of therapy initiation on outcomes, research has shown that the sooner an intervention takes place, the higher the impact on the altered brain circuity of children with ASDs, resulting in positive outcomes [72,73]. In addition, the predictive value of cognitive ability at pre-treatment on participants’ performance is not yet clear [21,72]. Of course, different researchers have used different inclusion criteria to attempt to account for the limitations and controversies that the previous ASD research presents. On the other hand, these controversies may highlight the need for more targeted interventions in response to age of treatment onset and cognitive ability at baseline [74,75]. Improvements in daily living skills have been previously associated with increased age, higher developmental quotient and lower symptom severity, while children with lower IQs and more severe symptoms have shown slower daily living skills gains. Caregivers and treatment providers may need to adjust their interventions according each child’s developmental and autism level [76,77,78]. On the other hand, motor skills are an essential component for social communication and social engagement, since they influence how someone responds to social stimuli. Gestures and facial expressions rely on motor function and are crucial for the communication and social engagement of children with ASDs. Motor skill deficits are common for people with ASDs and influence how people with ASDs interact and perceive communication with other people and their environment. For this reason, in recent years, research has been focused on investigating the mechanisms of motor skills and the impact that motor skills interventions can have on ASD-related symptoms [79,80,81].

Lately, more and more studies have highlighted the need for parent involvement and parent training. When parents are actively involved, either as main intervention providers or as co-therapists, intervention is associated with more positive outcomes for children with ASDs. Current evidence suggests that effective skills need to be intensively practiced in everyday life so that an intervention can be effective. Therefore, interventions in which families are trained to work daily on the skills that children with ASDs need to acquire may be the most promising. In order to be applicable and to maximize the effects, such interventions should be individualized, considering the needs of the child, family life, and their interactions [82,83].

Although our subgroup analysis failed to demonstrate any mediating effect of the duration and intensity of interventions on outcomes, it is strongly recommended to compare the early interventions based on their intensity and duration and not just on their manuals. Given the fact that the majority of the control groups in the current meta-analysis were assigned to TAU, it should be noted that, particularly in the case of very low and of high-intensity experimental interventions, the TAU group should be receiving at least as much intervention as the experimental groups so that the comparisons are meaningful. This was not always the case. Additionally, some studies, such as those involving very short-term interventions (e.g., 12 weeks), were not designed to examine whether major changes occur in cognition and language as a result of treatment but rather whether intervention can differentially alter or accelerate development in specific skills in ASDs. However, by including these studies, we wanted to see whether this acceleration could indirectly influence more comprehensive skills such as cognition, language and adaptive behavior as distal variables [2]. Additionally, early intervention may initiate a cascade of developmental events that lead to an altered brain circuity, resulting in better developmental outcomes [65]. For this reason, we performed a meta-analysis of follow-up data. According to our analysis, there is no evidence that experimental groups perform better than control groups after the course of the intervention. However, follow-up data were not reported consistently, so this analysis can be considered unstable due to the limited number of available studies. Another possible explanation for the absence of effects is that standardized measures are not always sensitive to the kinds of change activated by treatment for children with ASDs since they measure molar aspects of behavior and fail to detect the acquisition of specific skills [2].

The current study has certain limitations. One of the main limitations was the high variability of the included studies in the participants’ age and the targeted outcomes, as reported above. The provided interventions also varied regarding their duration, intensity, and structural elements. The combination of outcomes in statistical analysis could have diluted or masked the isolated gains that might have occurred due to the intervention. In children with a complex, multi-system disorder such as an ASDs, even those isolated gains are noteworthy. Furthermore, the current meta-analysis included data from pilot studies, such as those of Divan et al., (2019) [36], Drew et al., (2002) [37], Vernon et al., (2019) [63], and Reitzel et al., (2013) [50], that did not have a large number of participants. Additionally, many participants across all studies received other kind of therapies apart from the examined interventions during the study periods, and this could have exerted some influence on the results.

On the other hand, apart from its limitations, the current meta-analysis has many strengths. The first and most important is that this meta-analysis reports outcomes based only on RCTs, so many study quality parameters, such as the adequate randomization procedure, the existence of a control condition, the comparability of the groups at baseline, and the use of standardized measurements were fulfilled across all studies. Additionally, following the quality assessment of the included studies, those assessed as having a high risk of bias in the measurement of the outcome were excluded from the analysis, so it can be assumed that the outcomes ere accurate and valid. Moreover, using the formula suggested by Borenstein et al., (2021) [29], it was possible to combine all outcomes for the same variable across studies using a valid method to calculate the variances of the multiple outcomes so that each study was appropriately weighted in the final analysis. In this way, we included all the available information derived from different tests and scales for each single variable.

## 5. Conclusions

In conclusion, the current meta-analysis has demonstrated that although early intervention generally might not lead to positive outcomes for cognitive ability, language, communication and socialization for children with ASDs, these results should be interpreted with caution considering the great variability in participant and intervention characteristics. Perhaps it is not that the intervention itself is ineffective but that researchers should consider (a) the need for participants to attain sufficient intervention dosage, since previous literature has yielded positive outcomes for high-intensity interventions; (b) the need to design or use more sensitive measures than standardized measures; and (c) the need to examine longer-term effects through follow-up studies, since early intervention programs initiate a circuit of developmental events and children who received early intervention may continue to progress well years after the initial intervention. On the other hand, we identified significant positive effects of early intervention on daily living and motor skills, which have implications for everyday life and social communication. We recommend that future research should focus on creating more specific intervention groups using participants with comparable cognitive ability at baseline and a smaller age range in order to explore whether specific subgroups of children with ASDs respond better to early interventions than others. Additionally, we underline the need for future studies that meet the quality research standards in order to draw more valid and accurate conclusions.

## Figures and Tables

**Figure 1 jcm-11-05100-f001:**
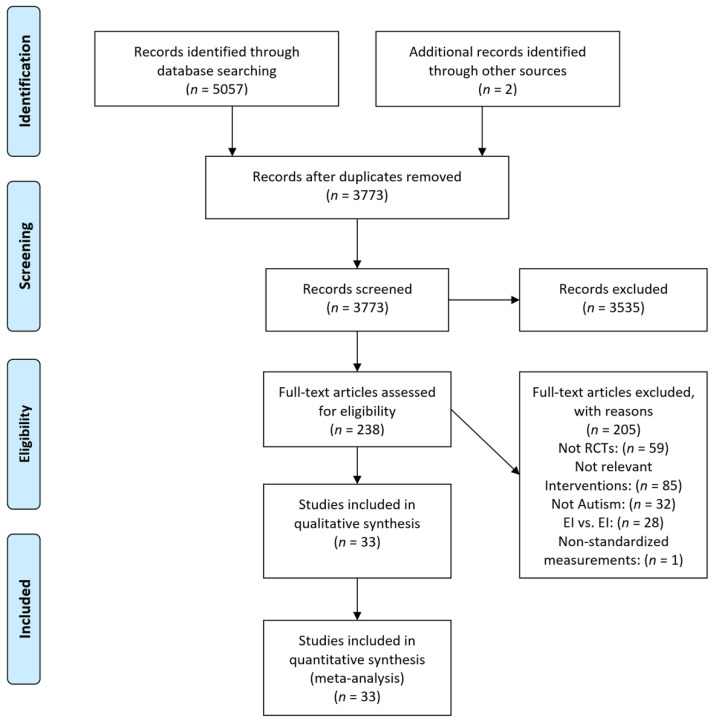
Flow diagram of study selection.

**Figure 2 jcm-11-05100-f002:**
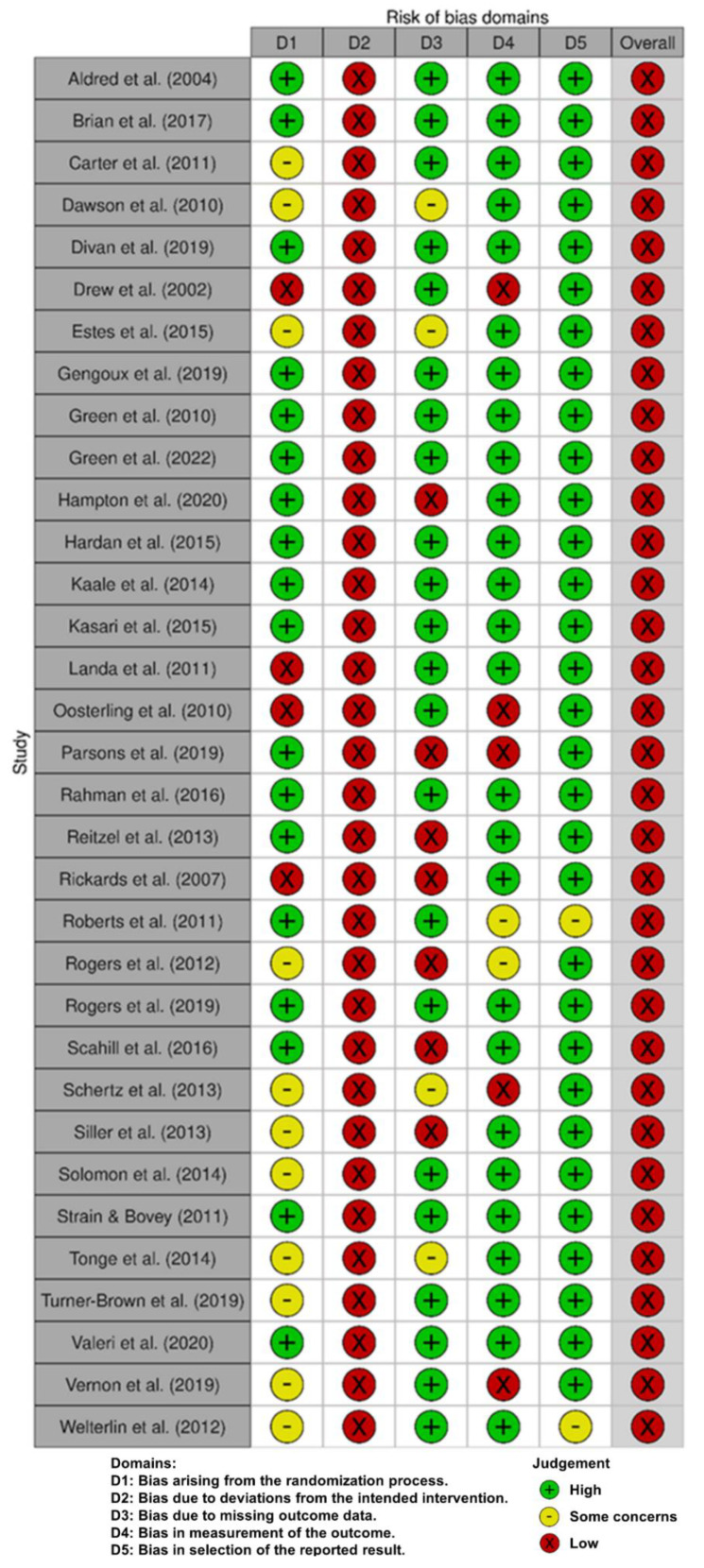
Risk of bias assessment [27,28,29,30,31,32,33,34,35,36,37,38,39,40,41,42,43,44,45,46,47,48,49,50,51,52,53,54,55,56,57,58,59].

**Figure 3 jcm-11-05100-f003:**
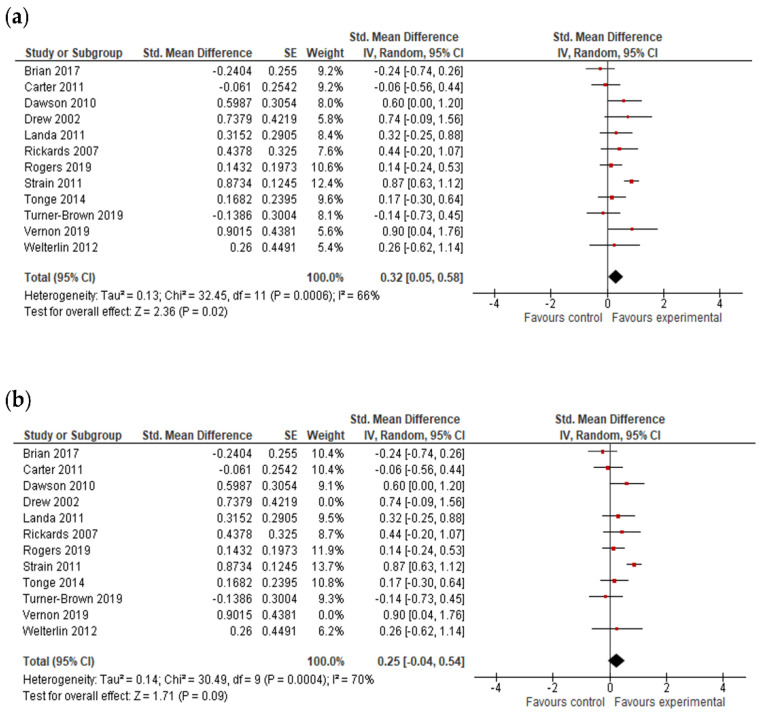
Forest plots for cognitive ability results. (**a**) Overall effect [28,29,30,32,41,46,49,54,55,56,58,59]. (**b**) Results after exclusion of studies with no blinding of outcome assessment [28,29,30,41,46,49,54,55,56,58,59].

**Figure 4 jcm-11-05100-f004:**
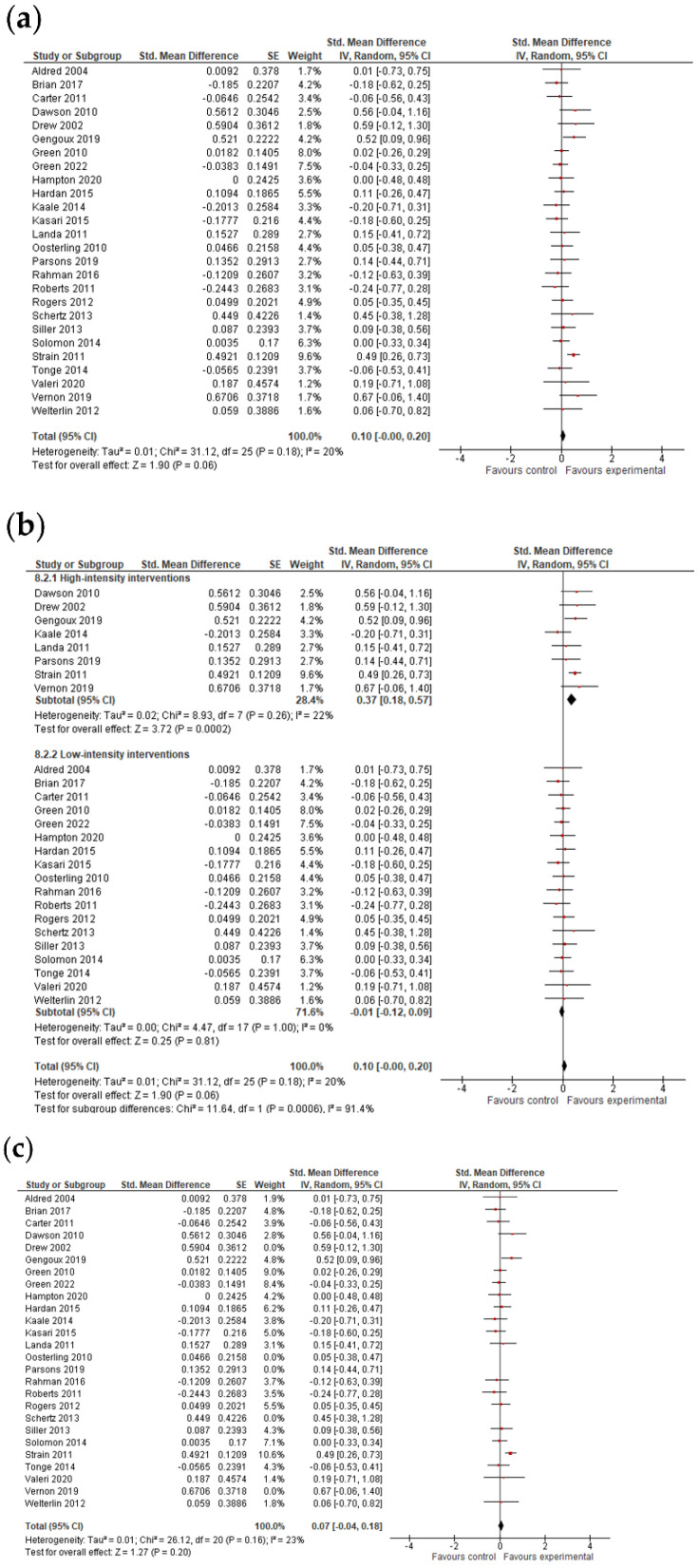
Forest plots for expressive language results. (**a**) Overall effect [27,28,29,30,32,34,35,36,37,38,39,40,41,42,43,44,47,48,51,52,53,54,55,57,58,59]. (**b**) Subgroup analysis: intensity of intervention. (**c**) Results after exclusion of studies with no blinding of outcome assessment [27,28,29,30,34,35,36,37,38,39,40,41,44,47,48,52,53,54,55,57,59].

**Figure 5 jcm-11-05100-f005:**
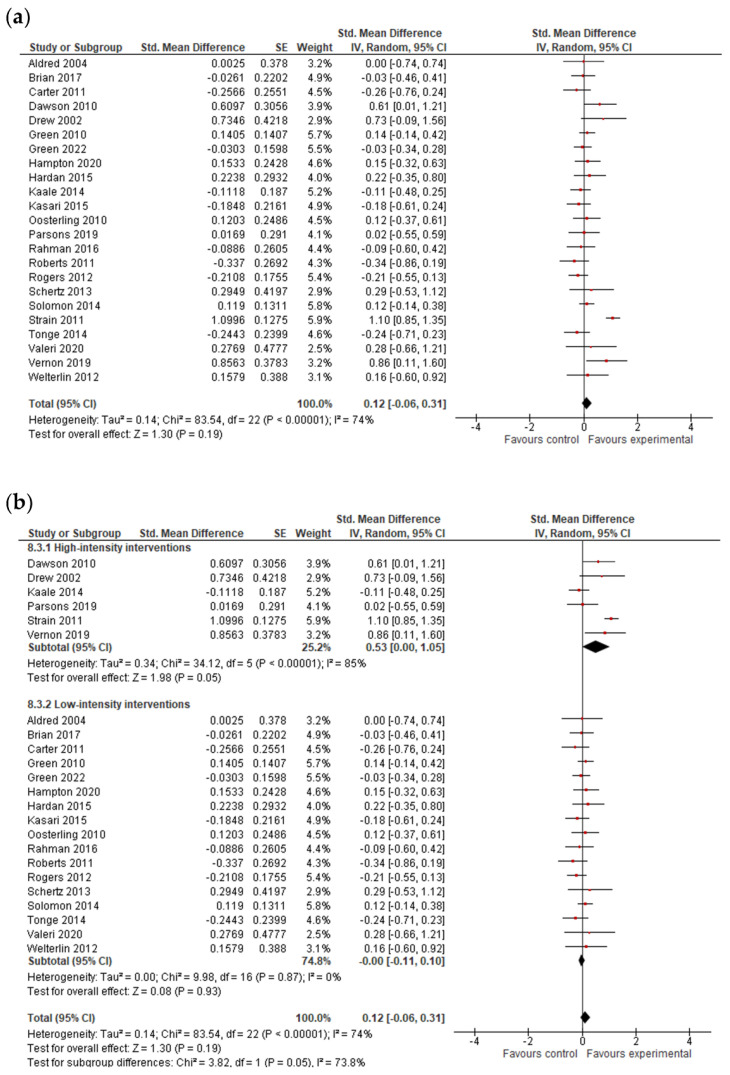
Forest plots for receptive language results. (**a**) Overall effect [27,28,29,30,32,35,36,37,38,39,40,42,43,44,47,48,51,53,54,55,57,58,59]. (**b**) Subgroup analysis: intensity of intervention.

**Figure 6 jcm-11-05100-f006:**
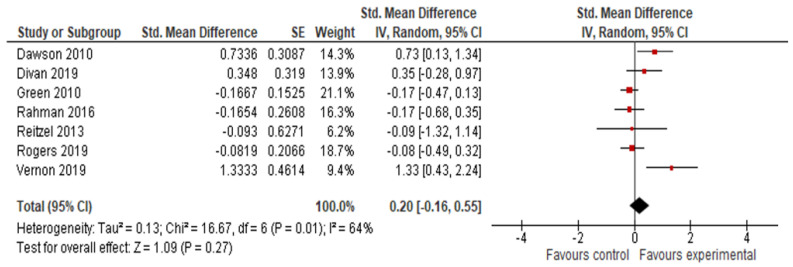
Forest plot for adaptive behavior composite. Overall effect [30,31,35,44,45,49,58].

**Figure 7 jcm-11-05100-f007:**
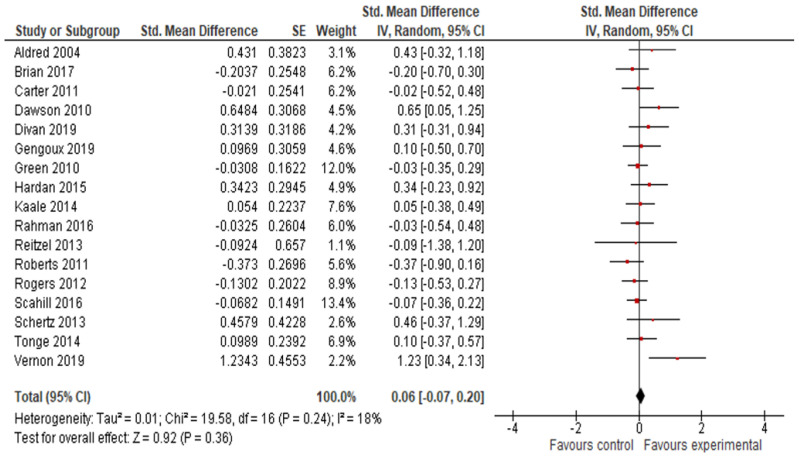
Forest plot for communication results. Overall effect [27,28,29,30,31,34,35,38,39,44,45,47,48,50,51,55,58].

**Figure 8 jcm-11-05100-f008:**
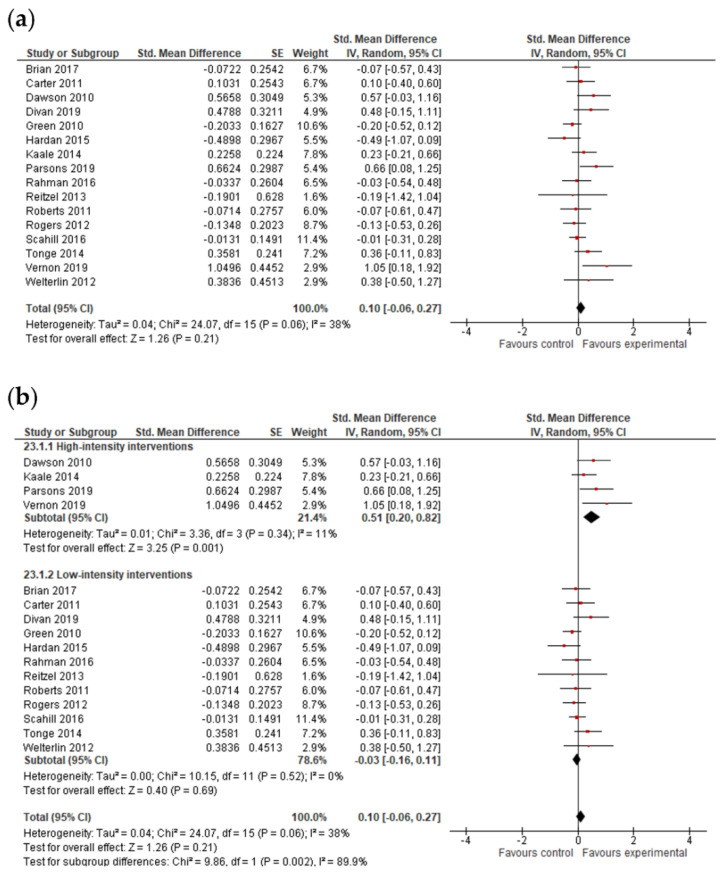
Forest plots for socialization results. (**a**) Overall effect [28,29,30,31,35,38,39,43,44,45,47,48,50,55,58,59]. (**b**) Subgroup analysis: intensity of intervention.

**Figure 9 jcm-11-05100-f009:**
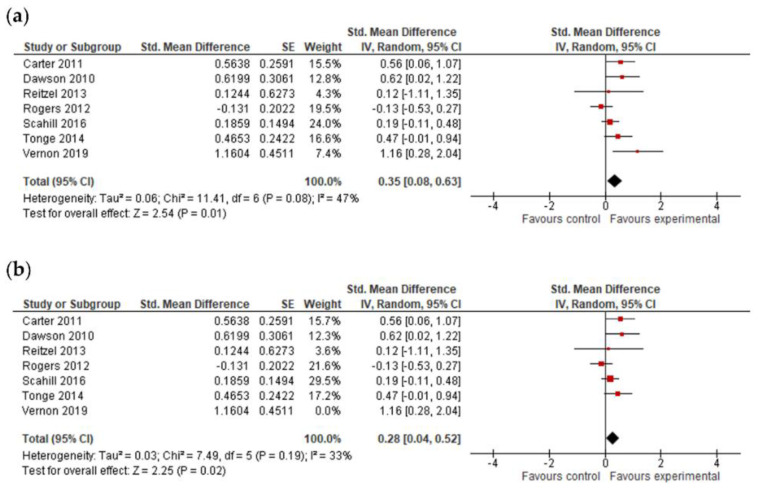
Forest plots for daily living skills. (**a**) Overall effect [29,30,45,48,50,55,58]. (**b**) Results after exclusion of studies with no blinding of outcome assessment [29,30,45,48,50,55].

**Figure 10 jcm-11-05100-f010:**
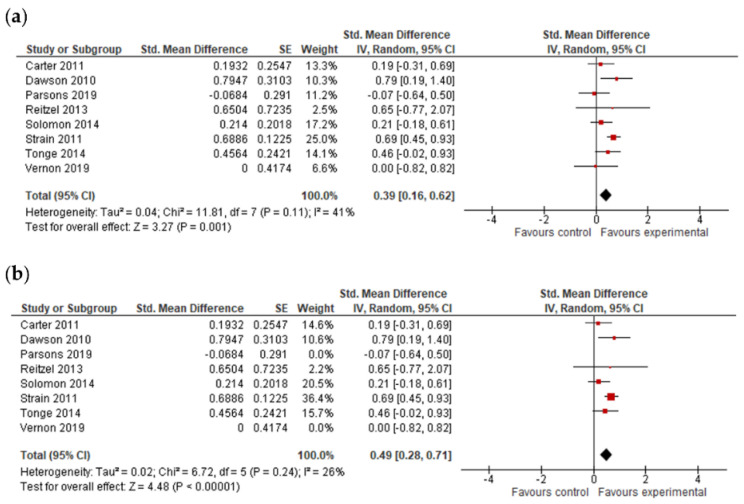
Forest plots for motor skills. (**a**) Overall effect [29,30,43,45,53,54,55,58]. (**b**) Results after exclusion of studies with no blinding of outcome assessment [29,30,45,53,54,55].

**Figure 11 jcm-11-05100-f011:**
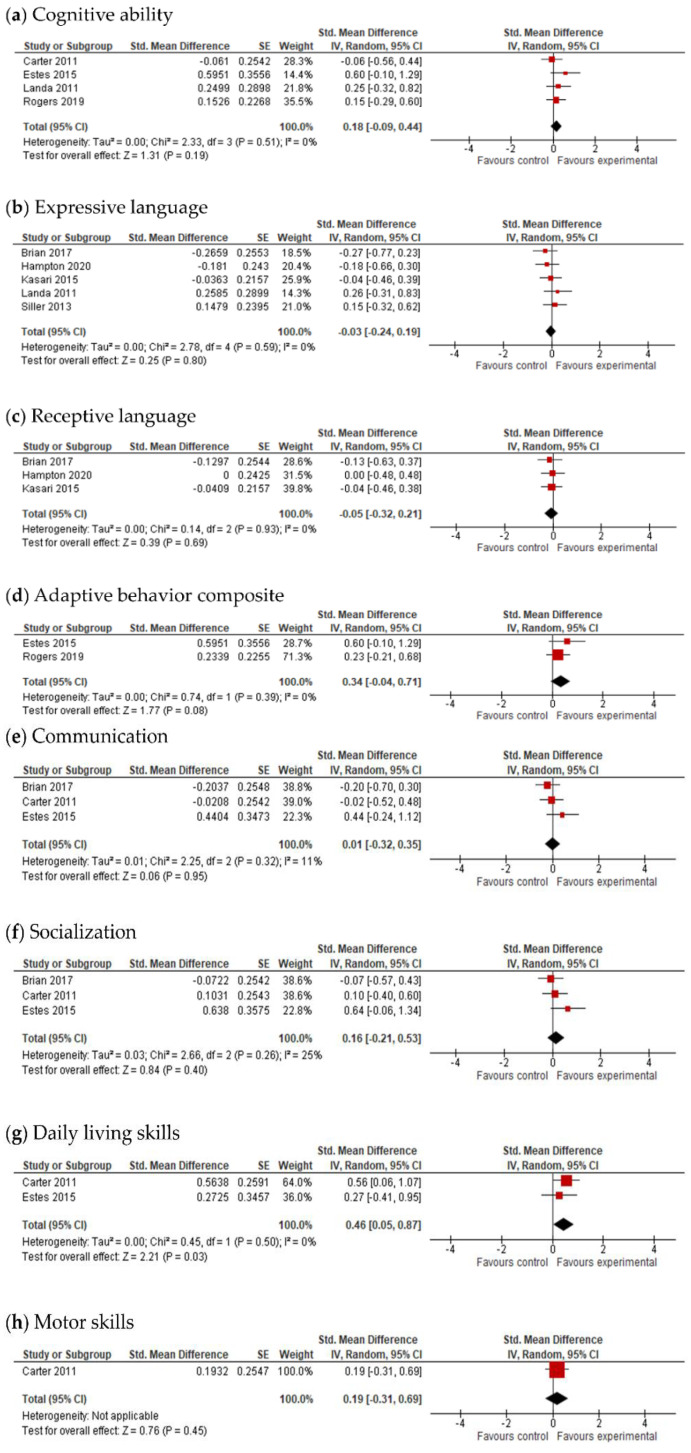
Forest plots for follow-up data. (**a**) Cognitive ability [29,33,41,49]. (**b**) Expressive language [28,37,40,41,52]. (**c**) Receptive language [28,37,40]. (**d**) Adaptive behavior composite [33,49]. (**e**) Communication [28,29,33]. (**f**) Socialization [28,29,33]. (**g**) Daily living skills [29,33]. (**h**) Motor skills [29].

**Figure 12 jcm-11-05100-f012:**
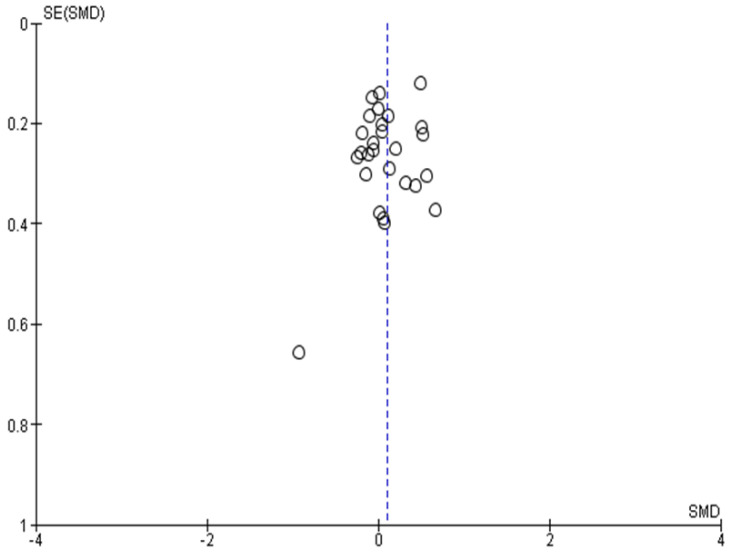
Funnel plot for publication bias.

**Table 1 jcm-11-05100-t001:** Characteristics of the included studies.

Study	Participants *n* (% Males), Mean Age (Age Range)	Experiment Group Intervention	Duration and Intensity of Intervention	Comparison Condition	Duration	Intervention Providers	Setting
Aldred et al. (2004) [27]	*n* = 28 (89.29%, males); mean age: 49.5 (24–71)	Social Communication Intervention	6 months with monthly sessions followed by another 6 months of 2-monthly consolidation sessions plus 30 min daily parent–child interaction	TAU	12 months	Professionals and parents	Individual
Brian et al. (2017) [28]	*n* = 62 (75.8%, males); mean age: 25.26 (16–30)	Social ABCs	12 weeks of 1.5 h home visits with tapering intensity (week 1: 3 visits; week 2: 2 visits; weeks 3–8: 1 visit/week; weeks 10 and 12: 1 “booster” visit/week; weeks 9 and 11: check-in phone call)	TAU: 3 months: up to 1 h/week of “other” direct therapy	3 months	Parents	Individual
Carter et al. (2011) [29]	*n* = 62 (82.26% males); mean age: 20.25 (15–25)	Hanen’s More Than Words (HMTW)	8 group parent sessions of 2.5 h and three in-home individualized parent–child sessions of 1 h	No treatment	3.5 months	Parents	Individual and group
Dawson et al. (2010) [30]	*n* = 48 (77.1% males); mean age: 23.5 (18–30)	The Early Start Denver Model (ESDM)	15.2 h therapist-delivered and 16.3 h/week parent-delivered therapy, 5 days/week	TAU: 9.1 h of individual therapy and an average of 9.3 h/week of group interventions for 2 years	2 years	Parents	Individual
Divan et al. (2019) [31]	*n* = 40 (87.5% males); mean age: 64 (27–105)	Parent mediated intervention for Autism Spectrum Disorder Plus (PASS Plus)	12 fortnightly home-based session between the parent and the lay health worker after a 10 min period of play between the parent and the child	TAU	6 months	Parents	Individual
Drew et al. (2002) [32]	*n* = 24 (79.17% males); mean age: 22.5 (-)	Parent training intervention with a focus on the development of joint attention skills and joint action routines	3 h parent sessions every 6 weeks	TAU: 3 months: 32.9 h/week	12 months	Parents	Individual
Estes et al. (2015), follow-up of Dawson et al. (2010) [33]	*n* = 39 (77% males); mean age: 2.9	The Early Start Denver Model (ESDM)	15.2 h therapist-delivered and 16.3 h/week parent-delivered therapy, 5 days/week	TAU: 9.1 h of individual therapy and an average of 9.3 h/week of group interventions for 2 years	2 years	Parents	Individual
Gengoux et al. (2019) [34]	*n* = 43 (88.4% males); mean age: 48.4 (24–60)	PRT-P	Weeks 1 to 12: weekly 60 min parent training sessions and 10 h per week of clinician delivered in-home treatment for children. Weeks 12 to 24: monthly 60 min parent training sessions and 5 h per week of in-home treatment for children	Waitlist and stable community treatments	24 weeks	Professionals and parents	Individual
Green et al. (2010) [35]	*n* = 152 (90.79% males); mean age: 45 (24–60)	Parent-mediated communication-focused treatment in children with autism (PACT)	2 h clinic sessions every 2 weeks for 6 months followed by monthly booster sessions for 6 months, plus 30 min of daily home practice	TAU	13 months	Parents	Individual
Green et al. (2022) [36]	*n* = 248 (79.4% males); mean age: 63 (24–132)	Paediatric Autism Communication Therapy-Generalised (PACT-G)	12 intervention sessions over 6 months at home plus 12 sessions over 6 months, again with 50% remote delivery	TAU	6 months	Professionals and parents	Individual
Hampton et al. (2020) [37]	*n* = 73 (79% males); mean age: 43 (36–60)	Caregiver training, Discrete Trial Teaching, and JASP + EMT + SGD	36 sessions in the clinic and at home, 45–60 min per session	TAU	4 months	Professionals and parents	Individual
Hardan et al. (2015) [38]	*n* = 47 (75% males); mean age: 49.2 (24–84)	Pivotal Response Treatment (PRT)	1 session of 90 min/week	Psychoeducation: 12 weeks: 1 session of 60 min/week	12 weeks	Parents	Individual
Kaale et al. (2014) [39]	*n* = 61 (78.7% males); mean age: 48.8 (24–60)	Social communication treatment	2 daily 20 min sessions, including 5 min of table-top training and 15 min of floor play	TAU	8 weeks	Teachers	Group
Kasari et al. (2015) [40]	*n* = 86 (81.4% males); mean age: 31.5	Joint Attention, Symbolic Play, Engagement and Regulation (JASPER)	2 sessions of 30 min per week	Psychoeducation: 1 h per week	10 weeks	Professionals and parents	Individual
Landa et al. (2011) [41]	*n* = 48 (88.3% males); mean age: 28.7 (21–33)	Interpersonal Synchrony	10 h per week in classroom, student-to-teacher ratio, schedule, home-based parent training (1.5 h per month), parent education (38 h), plus supplementary curriculum targeting socially engaged imitation, joint attention, and affect sharing	Non-Interpersonal Synchrony: 10 h per week in classroom, student-to-teacher ratio, schedule, home-based parent training (1.5 h per month), parent education (38 h)	6 months	Teachers, Parents	Group, Individual
Oosterling et al. (2010) [42]	*n* = 65 (77.61% males); mean age: 34.32 (<12–42)	Joint attention and language skills stimulation	2 h sessions with parents, 3 h home visits every 6 weeks during the first year. In the second year, 3-month intervals between home visits	TAU	2 years	Parents	Individual
Parsons et al. (2019) [43]	*n* = 59 (81.4% males); mean age: 62.6 (24–72)	TOBY app targeting visual motor, imitation, language and social parameters	At least 20 min on the TOBY app daily for 3 months using an iPad	TAU	3 months	-	Individual
Rahman et al. (2016) [44]	*n* = 65 (81.5% males); mean age: 64.5 24–108)	PASS (plus TAU)	1 h sessions every 2 weeks for 6 months	TAU	6 months	Professionals and parents	Individual
Reitzel et al. (2013) [45]	*n* = 11 (-); mean age: 58.5 (38–82)	Functional Behaviour Skills Training program (FBST)	30 min parents-only training sessions, a simultaneous children’s activity session, and a 90 min combined children’s and parents’ training session	TAU	4 months	Professionals and parents	Individual
Rickards et al. (2007) [46]	*n* = 59 (79.7% males); mean age: 43.87 (36–60)	Home-based Program (in addition to a center-based program)	1 and 1½ h during school terms over a 12-month period plus 5 h spread over two weekly sessions during school terms	Center-based program: 5 h spread over two sessions weekly during school terms	12 months	Professionals	Individual
Roberts et al. (2011) [47]	*n* = 56 (90.5% males); mean age: 42.6 (26.3–60.3)	Building Blocks home-based	Visit for 2 h once a fortnight over a 40-week period (20 sessions maximum)	Waitlist	1 year	Parents	Individual
Rogers et al. (2012) [48]	*n* = 98 (77.55% males); mean age: 20.98 (12–24)	Brief Early Start Denver Model (P-ESDM) Parent-based Intervention	1 session of 1 h/week	TAU	12 weeks	Parents	Individual
Rogers et al. (2019) [49]	*n* = 118 (78% males); mean age: 21.02 (14–24)	Early Start Denver Model (ESDM)	3 months of weekly parent coaching followed by 24 months of 15 h per week (on average) 1:1 treatment weekly on average in homes or daycare settings from supervised therapy assistants while parents received 4 h of coaching monthly from a certified ESDM therapist	TAU	27 months	Professionals and parents	Individual
Scahill et al. (2016) [50]	*n* = 180 (87.7% males); mean age: 4.75 (36–83)	Parent training (PT)	Eleven 60-to-90-minute core sessions, up to 2 optional sessions, and a home visit over 16 weeks, as well as a home visit and 2 telephone booster sessions between weeks 16 and 24	Structured parent education program (PEP): twelve 60-to-90-minute individually administered sessions and 1 home visit over 24 weeks	24 weeks	Parents	Individual
Schertz et al. (2013) [51]	*n* = 23 (-); mean age: 26.11 (<30)	Joint Attention Mediated Learning (JAML)	15 home visits included 10 min parent–child interaction plus 30 min daily parent–child interaction	TAU	7 months	Parents	Individual
Siller et al. (2013) [52]	*n* = 70 (91% males); mean age: 57.1 (32–82)	Focused Playtime Intervention (FPI)	1 session per week for 12 weeks, 90 min per session	PAC	12 weeks	Parents	Individual
Solomon et al. (2014) [53]	*n* = 128 (78.91% males); mean age: 50.19 (32–71)	PLAY Project Home Consultation program (PLAY)	3 h home visits/month	TAU: 2 h/week	1 year	Parents	Individual
Strain and Bovey (2011) [54]	*n* = 294; mean age: 50.33	LEAP intervention (Learning Experiences and Alternative Program for Preschoolers and Their Parents)	2.75–3 h per day, 5 days per week	Intervention manuals and related written materials to preschool staff: 2.75–3 h per day, 5 days per week	2 years	Professionals and parents	Group
Tonge et al. (2014) [55]	*n* = 70 (82.86% males); mean age: 46.56 (23–70)	Education and behavior management skills for Pre-schoolers with Autism (PEBM)	Ten 90-minute small group (4–5 families) sessions alternated with ten 60-minute individual family sessions over a 20-week period.	TAU	20 weeks	Parents	Individual and group
Turner-Brown et al. (2019) [56]	*n* = 49 (85.7% males); mean age: 29.6 (17–35)	Family Implemented TEACCH for Toddlers (FITT)	Twenty 90-minute in-home sessions where the FITT coach works directly with the family and toddler and 4 parent group sessions.	TAU	6 months	Parents and professionals	Individual and group
Valeri et al. (2020) [57]	*n* = 34 (79% males); mean age: 48.3 (24–132)	Cooperative parent-mediated therapy (CPMT) plus low-intensity psychosocial intervention (LPI)	15 sessions of 60 min. 12 core sessions, 1 per week, were delivered in the first 3 months, followed by 3 monthly booster sessions plus 4 h LIP per week	Low-intensity psychosocial intervention (LPI)	6 months	Parents and professionals	Individual
Vernon et al. (2019) [58]	*n* = 23 (87% males); mean age: 35.13 (18–56)	Pivotal Response Intervention for Social Motivation (PRISM)	10 h a week of intervention: 8 h of one-on-one clinician-implemented treatment and 2 h of parent education in the intervention strategies with the child present	TAU	6 months	Parents and professionals	Individual
Welterlin et al. (2012) [59]	*n* = 20 (90% males); mean age: 30.5 (24–39)	Home TEACCHing Program for Toddlers with Autism	1.5 h per week for 12 sessions plus 30 min of parents’ psychoeducation	Waitlist	12 weeks	Parents	Individual

## Data Availability

All data are available in the main text or the extended data. The search protocols and datasets generated and/or analyzed during the current study are available from the corresponding author on reasonable request.

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
