# Peer review of "The Efficacy of Early Interventions for Children with Autism Spectrum Disorders: A Systematic Review and Meta-Analysis"

_jcm, 2022, doi:10.3390/jcm11175100_

Round 1

Reviewer 1 Report

1.      Keywords are recommended to add at least five keywords for more searchable literature if published later.

2.      In line 69, “The aim of the current study is to…”. The article type is Review, but in the objective is the study? Something wired, if it is a Review article, it should be summarized or similar terms. Review is not studied!

3.      What is the novel of the present article, the are many similar published review articles presenting systematic review and meta-analysis of the delivered topic, such as:

Efficacy of interventions to improve feeding difficulties in children with autism spectrum disorders: a systematic review and meta‐analysis. Child: care, health and development. 2014. https://doi.org/10.1111/cch.12157

The authors need to highlight more advance their novel in the present review article. This issue is mandatory for the authors.

4.      Please make a paragraph of at least three sentences consisting of one topic sentence and following with a supporting sentence for making a solid explanation. For example, is in lines 80-81 make a hanging sentence. So wired if one paragraph is only consisting of one sentence as present form. Revise it.

5.      In line 28-31 of the Introduction section, the authors need to adopt more literature for explaining the sensory-based intervention approach, one of the is sensory-based intervention using a hug machine with deep pressure. The suggested literature published by MDPI should be adopted as follows: Physiological Effect of Deep Pressure in Reducing Anxiety of Children with ASD during Traveling: A Public Transportation Setting. Bioengineering 2022, 9, 157. https://doi.org/10.3390/bioengineering9040157

6.      In the literature searching strategy, please used three main databases, there are Scopus, Web of Science, and PubMed. The present source used by the authors is not appropriate.

7.      In line 144, what is the Revman software version? Is should be detailed.

8.      Further research needs to be explained in the conclusion section.

9.      Please recheck again for the present manuscript that has been followed by Prisma 2020.

10.   Grammatical error and language style needs to be solved. The authors need to proofread the present manuscript. Alternatively, MDPI English language service would be used.

11.   Please make sure the authors have followed the Journal of Clinical Medicine format properly.

12.   Enrich the literature from the last 5 years published that showed the present study is the least research. Literature published by MDPI is strongly encouraged.

Reviewer 2 Report

The authors investigated the efficacy of early interventions for children with Autism Spectrum Disorders (ASD). Meta-analysis included studies investigating cognitive ability, expressive language, receptive language, communication, adaptive behavior composite, daily living skills and motor skills in Randomized Controlled Trials (RCTs). The analysis was performed by two independent authors using PRISMA. The intervention duration and intensity were also evaluated. Positive outcomes were found in cognitive ability, daily living skills, and motor skills. The variability in participants´ and intervention characteristics but no further analyses directed to ASD subtypes were performed. Since early interventions comprise a group of different types of approaches, the results are very limited and request refinement for publication.

The age range of the sample was 1-9 years (12-109 months, but actually Valeri et al. 2020 study age range 24-132 mo???) and the question raised, could better targeted intervention be more valuable?

Since the early intervention improved very basic skills such as motor skills, it would be interesting to define this effect more in detail in the view of problems particularly in ASD.

The duration, intensity, and structural elements of the interventions varied a lot making the meta-analysis noncomprehensive and results unnecessary as presented. As discussed, preliminary studies are included to the meta-analysis. Could the selection be better defined?

The manuscript is well written and integrated to the current situation in the research field. In introduction there is typos in Developmental and Naturalistic Developmental Behavioral Interventions (NCBI) and abbreviation RCT is not introduced.

Round 2

Reviewer 1 Report

This manuscript is recommended for publication in the Journal of Clinical Medicine.